# Innate Immune Response to Viral Vectors in Gene Therapy

**DOI:** 10.3390/v15091801

**Published:** 2023-08-24

**Authors:** Yixuan Wang, Wenwei Shao

**Affiliations:** Academy of Medical Engineering and Translational Medicine, Medical College, Tianjin University, Tianjin 300072, China; 2021235075@tju.edu.cn

**Keywords:** innate immune, adeno-associated virus, lentivirus, adenovirus

## Abstract

Viral vectors play a pivotal role in the field of gene therapy, with several related drugs having already gained clinical approval from the EMA and FDA. However, numerous viral gene therapy vectors are currently undergoing pre-clinical research or participating in clinical trials. Despite advancements, the innate response remains a significant barrier impeding the clinical development of viral gene therapy. The innate immune response to viral gene therapy vectors and transgenes is still an important reason hindering its clinical development. Extensive studies have demonstrated that different DNA and RNA sensors can detect adenoviruses, adeno-associated viruses, and lentiviruses, thereby activating various innate immune pathways such as Toll-like receptor (TLR), cyclic GMP-AMP synthase–stimulator of interferon genes (cGAS-STING), and retinoic acid-inducible gene I–mitochondrial antiviral signaling protein (RLR-MAVS). This review focuses on elucidating the mechanisms underlying the innate immune response induced by three widely utilized viral vectors: adenovirus, adeno-associated virus, and lentivirus, as well as the strategies employed to circumvent innate immunity.

## 1. Introduction

Gene therapy exhibits considerable potential for advancement in the 21st century. To achieve therapeutic effects, gene therapy relies on vectors to deliver the required genes to target cells. These vectors can be classified into two main categories: viral and non-viral types. Presently, viral vectors enjoy extensive utilization within the field of gene therapy due to their widespread adoption and effectiveness.

Currently, prevalent viral vectors in gene therapy encompass adenovirus (Ad), adeno-associated virus (AAV), and lentivirus (LV) (Figure 1). Human Ad (HAdV) was first isolated from human adenoid tissue cultured by Wallace Rowe and his colleagues in 1953. It was considered to be a gene therapy vector around the 1980s [1]. In 1992, the first successful utilization of Ad as a vector for gene therapy was reported. AAV, initially discovered in the mid-1960s, served as a vector for ex vivo gene transduction in 1984. It was first employed to treat patients with cystic fibrosis in 1995 and exhibited encouraging therapeutic outcomes in patients with Leber’s congenital amaurosis in 2008. Following that, in 2012, Glybera (Table 1) became the pioneering AAV gene therapy drug to be approved by European Medicines Agency (EMA) [2]. LV, derived from the human immunodeficiency virus (HIV), serves as a viral vector. Upon gene transduction by the vector, if the cell can persist over an extended period, the transgene integrates into the host cell’s genome, thereby facilitating enduring transgene expression [3]. As a result of the integration capability exhibited by LV vector, it is prioritized as the primary option in gene therapy.

Innate immunity functions as the initial barrier against foreign intrusions and assumes a pivotal role in host protection [4]. Operating through swift and non-specific reactions, it often triggers cytokine responses within an hour. Given the diverse pathways through which viruses can infiltrate cells, the host engages in a sequence of biological processes aimed at countering viral entry. Antigen-presenting cells, including plasmacytoid dendritic cells (DCs), conventional DCs, macrophages, and B cells, emerge as significant generators of antiviral cytokines [5]. When encountering viruses, the innate immune system recognizes these viral entities as foreign by employing pattern-recognition receptors (PRRs). These PRRs have the capacity to identify distinctive structures inherent to non-self agents, commonly denoted as pathogen-associated molecular patterns (PAMPs) [6]. Subsequent to this recognition, PRRs can instigate the activation of an array of intracellular signaling pathways, potentially leading to the initiation of innate immune responses [7]. Among these receptors, Toll-like receptors (TLRs) expressed on cellular membranes, retinoic acid-inducible gene-I (RIG-I)-like receptors (RLRs), nucleotide oligomerization domain (NOD)-like receptors (NLRs), C-type lectin receptors (CLRs), and a spectrum of intracellular DNA sensors such as cyclic GMP-AMP synthase (cGAS) play pivotal roles in transducing these immune-initiating signals [8]. Upon detection of viral PAMPs, PRRs become activated, subsequently engaging NF-κB and interferon regulatory transcription factors IRF3 and IRF7. This orchestrated activation cascade orchestrates the transcription of proinflammatory cytokines and type I interferons within the infected cells. Consequent to their synthesis, these interferons bind to their designated IFN-I receptors, thereby initiating the expression of IFN-stimulated genes (ISGs). These molecular cues serve to attract effector lymphocytes, impede viral transduction, and propel the elimination of transduced cells via the instigation of an adaptive immune response.

Adenovirus disassembly in the endosomes, accompanied by the exposure of viral DNA (Table 2) in plasmacytoid dendritic cells, precipitates the initiation of the TLR9/MyD88 axis. This process subsequently leads to IRF7-dependent transcription of IFN-I genes. Alternatively, the perception of viral DNA in the cytoplasm provokes TLR-independent activation of type I interferon through the cGAS/STING/TBK1/IRF3 signaling cascade [9]. The manner of HAdV recognition is contingent upon cell type, involving both cytoplasmic and nuclear sensing mechanisms, thereby inciting an innate immune response to the infection. In human cell lines, the Ad genome triggers activation of the cytoplasmic cGAS-STING pathway, which in turn activates TBK1 and IRF3, ultimately culminating in the expression of IFN-β genes. The release of IL-1β induced by Ad hinges on the detection of the Ad genome by the DNA sensor TLR9. In murine models, Ad infection traverses plasmacytoid dendritic cells (pDCs). Notably, conventional dendritic cells and macrophages elicit substantial levels of type I interferon. Recognition of Ad in pDCs is mediated by TLR9, whereas non-pDC recognition of Ad is TLR9-independent and instead contingent upon the cytosolic sensing of viral DNA. In murine antigen-presenting cells (APCs), Ad infection triggers IRF6-mediated IFN responses via TLR-independent DNA sensing, partly reliant on the cGAS-STING pathway [10]. Several studies have attested to the activation of the RLR-MAVS signaling pathway during Ad virus infection. In Ad-transduced host cells, Ad-associated RNA (VA-RNA) engages RIG-I, thereby instigating the RLR-MAVS pathway and subsequently fostering the production of IFN-I [11].

DNA viruses, including AAV (Table 2), have the capability to undergo uncoating and discharge their genomes within the endosome. This action facilitates the engagement of TLR9, thereby instigating the initiation of the innate immune response. As AAV undergoes endosomal trafficking, there arises the potential for virion uncoating or partial genome exposure, which could consequently enable TLR9 detection of CpG motifs. This process transpires rapidly and consequently activates NF-κB, ultimately leading to the synthesis of proinflammatory cytokines [12]. Within the cellular cytosol, the DNA sensor cGAS can effectively bind double-stranded DNA, subsequently eliciting a signaling cascade that involves STING and TBK1. Notably, AAV ITRs exhibit promoter activity, thereby implying that the presence of both 5′- and 3′-ITRs within the AAV genome could potentially result in the generation of sense and antisense RNAs. These RNA molecules possess the capability to form double-stranded RNA intermediates within the cytoplasm. It is pertinent to mention that double-stranded RNA molecules are duly recognized by MDA5 and RIG-I, subsequently instigating the production of IFN-β [13].

HIV-1 (Table 2) infection triggers immune system responses through the recognition of various HIV-associated PAMPs by PRRs. Both cell membrane-expressed TLRs and endosomal TLRs recognize specific PAMPs in HIV, thus initiating the innate immune response. TLR2 and TLR4 present on the cell surface play roles in detecting HIV glycoprotein gp120, leading to NF-κB activation and subsequent production of inflammatory cytokines. Within phagocytic immune cells, TLR7/8 can identify single-stranded DNA (ssDNA), resulting in the secretion of IFN-α and inflammatory cytokines. The HIV virus generates intermediates that replicate within the cytoplasm, where the cytosolic HIV genomic RNA is recognized by RIG-I, prompting the expression of ISGs. Furthermore, beyond the viral genome, RIG-I in virus-infected macrophages can also detect newly synthesized viral mRNA post-viral replication, subsequently promoting ISG expression. As a retrovirus, HIV-1 produces a nucleic acid intermediate, double-stranded DNA (dsDNA), during reverse transcription. This dsDNA within the cytoplasm can be detected by cGAS, leading to the induction of IFN-β production [14].

This review summarizes the mechanisms of innate immune response and immune escape measures after the transduction of therapeutic genes by three viral vectors, Ad, AAV, and LV, which are widely used in gene therapy. Viruses inherently trigger innate immune responses; nevertheless, within this review, our concentration is solely on the innate responses provoked by genetically modified viral vectors utilized in gene therapy, along with strategies to counteract these innate immune reactions. Endeavors to circumvent immune responses in gene therapy can be classified into two primary categories: strategies aimed at shielding the vector and its transgene product from immune surveillance, and those designed to obscure the vector/transgene product from immune detection [5]. While RNA modification or vector alteration can curtail the likelihood of an innate immune response, they cannot entirely avert its occurrence. In practical application, the emergence of an innate immune response is an unavoidable outcome.

**Table 1 viruses-15-01801-t001:** Currently authorized viral vector drugs.

Date	Drugs	Regulatory Approval	Application	Vector
2003	Gendicine (recombinant human p53 adenovirus) [15]	China Food and Drug Administration (CFDA)	Head and neck squamous cell carcinoma (HNSCC)	Ad-p53
2006	Oncorine [16]	Chinese SFDA	Head and neck cancers, liver cancers, pancreatic cancers, cervical cancers, and other cancers	H101
2012	Alipogene tiparvovec (Glybera) (it was withdrawn from the market in 2017) [17]	EMA	Lipoprotein lipase (LPL) deficiency	AAV1-LPL
2017	Luxturna [18]	FDA	Leber congenital amaurosis caused by RPE65 mutations	AAV2-RPE65
2017	Kymriah (tisagenlecleucel) [19]	FDA	Acute lymphoblastic leukemia and diffuse large B-cell lymphoma	LV-CD19
2019	Zolgensma [20]	FDA	Spinal muscular atrophy (SMA)	scAAV9-SMN1
2020	Libmeldy [21]	EU	Metachromatic leukodystrophy (MLD)	SIN LV vector
2021	Elivaldogene autotemcel (Skysona, eli-cel) [22]	EU	Cerebral adrenoleukodystrophy (CALD)	LV. ABCD1
2021	Breyanzi (lisocabtagene maraleucel) [23]	FDA	Patients with relapsed or refractory large B-cell lymphomas	LV-CD19
2021	Abecma (idecabtagene vicleucel, ide-cel) [23]	FDA	Relapsed or refractory multiple myeloma (R/R MM)	LV-CD19
2022	Eladocagene exuparvovec (Upstaza) [24]	EMA	Human aromatic L-amino acid decarboxylase (AADC) deficiency	rAAV2-hAADC
2022	Roctavian (valoctocogene roxaparvovec) [25]	EMA	Severe hemophilia A [congenital factor VIII (FVIII) deficiency] in adults	BMN 270: AAV5 -hFVIII-SQ
2022	Adstiladrin [26]	FDA	Patients with NMIBC who do not respond to BCG	rAd-IFNα/Syn3
2022	Zynteglo (betibeglogene autotemcel, beti-cel) [27]	FDA	β-thalassemia patients	BB305 LV vector

**Table 2 viruses-15-01801-t002:** The differences between wild-type viruses and viral vectors.

Virus Type		Size	Genome Structure	Genome Type	Immunogenicity	Integration
Ad	Wild-type Virus	26–45 kb	ITR, φ, E1A, E1B, E2, E3, E4, L1–L5	dsDNA	High	Rarely
	Viral Vector	OAd: 3 kb; HDAd: 34 kb	ITR, φ, transgene			
AAV	Wild-type Virus	4.7 kb	ITR, Rep, Cap, AAP, MAAP	ssDNA	Low	Rarely
	Viral Vector	4.7 kb	ITR, transgene			
LV	Wild-type Virus	8–9 kb	LTR, gag, pol, env, rev, tat, vpr, vpu, vif, nef	ssRNA	Moderate	Random
	Viral Vector	<5 kb	1: gag, pol, rre, transgene; 2: rev, transgene			

## 2. Adenovirus Vector Therapy

### 2.1. Introduction to Adenovirus Vectors

#### 2.1.1. Principle of Gene Therapy with Adenoviral Vectors

Ad is an icosahedral virus with a diameter of 70–100 mm and no envelope capsids. Its genetic material is double-stranded DNA (dsDNA), with a length of 25–46 kb [28] and a 103 bp inverted terminal repeat (ITR) at each end of the genome, crucial for viral DNA replication (Figure 1). Ad can infect both dividing and non-dividing cells, is able to replicate in the infected nucleus with high transfection efficiencies and is not integrated into the host genome [29]. Nevertheless, Ad is not without limitations. It demonstrates heightened immunogenicity, leading to notable inflammatory responses. Additionally, it exhibits suboptimal infection efficiencies in certain cancer contexts. Ad is classified into seven types, A–G. To date, a cumulative count of 111 HAdV genotypes has been documented and categorized into seven species (A–G), all of which have the potential to infect individuals across various age groups, albeit only a limited subset leading to severe infections [30]. These vectors are mainly derived from Ad serotype 2 (Ad2) and Ad serotype 5 (Ad5) of species C, with Ad5 being the most widespread [31]. Different HAdV serotypes exhibit distinct tropism, capabilities, and clinical manifestations [32]. Among them, the HAdV-5 vector is currently the most used vector for cancer treatment. During adenovirus (Ad) infection, transcription occurs in distinct regions known as early (E), intermediate (I), and late (L) regions at different stages. The E region includes E1A, E1B, E2, E3, and E4 genes, primarily contributing to Ad replication. The L region comprises genes L1–L5, which are mainly responsible for coding structural and non-structural proteins. These proteins play essential roles in capsid formation, DNA packaging, and maturation of offspring Ad [33].

Ad5 can be categorized into two types based on their replication abilities: replication-deficient viruses and replication-competent oncolytic viruses (Table 2).

Over the last three decades, remarkable progress has been achieved in the manipulation of Ad genomes. The most recent iteration of Ad vector technology is represented by the helper-dependent Ad (HDAd), wherein all viral coding sequences have been excised from the genome, retaining solely the cis-acting ITRs and packaging sequences. This configuration offers a transgene capacity of up to 34 kb [34]. The E1-deficient adenovirus, being replication-defective, serves as an ideal shuttle vector for applications in gene therapy and vaccination [1].

Adenoviruses capable of conditionally activating the E1 gene for replication within tumor cells are termed conditionally replicating adenoviruses (CRAd), which are clinically referred to as oncolytic adenoviruses (OAds). The current adenovirus vectors utilized in clinical applications encompass four strategies to attain conditional replication. One approach involves governing E1A and adenovirus replication via cancer cell-specific promoters. These exogenous promoters are generally inserted into the E1A regulatory region, exploiting the scarcity or complete absence of tumor-specific promoters. This characteristic hinges on their capacity to induce robust expression of specific genes crucial for the malignant phenotype. The insertion of exogenous promoters predominantly occurs in the E1A regulatory region without substantial deletions. The remaining three methodologies predominantly entail modifications to the transcription units E1A and E1B [35]. Currently, the new generation of OAds is created through the deletion of 24 amino acids in the CR2 domain of the E1A protein, leading to the generation of the AdΔ24 vector. The CR2 domain binds to the retinoblastoma protein (pRb) and facilitates the release of S-phase activating transcription factors (E2F) essential for viral replication in normal cells. These vectors have demonstrated a remarkable combination of high replication efficiency and selectivity across diverse tumor cells. OAds necessitate the preservation of the majority of Ad genes to ensure efficient replication and cleavage function. Consequently, their transgene capacity is confined to approximately 3 kb.

#### 2.1.2. Application of Adenoviral Vectors in Gene Therapy

Currently, several vector gene therapy drugs for Ad have been certified (Table 1). In 2003, Gendicine was approved by the CFDA as a rAd-p53 drug for the treatment of HNSCC [15]. Most studies have combined rAd-p53 with other traditional therapies, and the results show that combination therapy is more effective than traditional treatment [36]. In 2006, Oncorine received approval from the State Food and Drug Administration for marketing [16]. It is primarily employed in the treatment of head and neck tumors, specifically nasopharyngeal carcinoma. The treatment regimen involves local injections combined with chemotherapy [37]. In 2022, Adstiladrin received FDA approval as a rAd-IFNα/Syn3 vector gene therapy for the treatment of adults with high-risk BCG non-muscle-invasive bladder cancer (NMIBC) [26]. In clinical studies, about 53% of patients achieved complete remission within 3 months of treatment [38].

For certain types of cancers, clinical trials have demonstrated that gene therapy using Ad vectors can effectively enhance therapeutic outcomes. Ad vectors have shown promise in the treatment of metastatic diseases, such as prostate cancer, where limited treatment. At present, a number of clinical trials utilizing Ad5 as a vector for localized prostate cancer treatment have exhibited favorable safety profiles and notable efficacy [37].

Tahir Muhammad et al. [39] conducted a study utilizing OAd packaged with Ad E1A/B gene-modified human mesenchymal stem cells, which could effectively suppress the growth of prostate cancer cells in mouse models. Furthermore, Tien V Nguyen et al. [40] observed minimal mouse injury upon administration of Ad657 compared to the widely used Ad5. This resulted in significant inhibition of tumor growth, prolonged survival time in mice, and notably improved therapeutic effects [40].

OAd vectors are also widely used in the treatment of liver cancers. Jian Meng et al. [41] conducted a study involving surgery combined with Oad administration in patients with small lung cancer. The gene therapy group received gene therapy prior to surgery, resulting in a notable increase in the overall survival rate of the gene therapy group and a significant reduction in postoperative recurrence probability. In vitro cell experiments have demonstrated the effectiveness of recombinant OAds, such as silica-coated OAds encoding anti-cancer gene [42], KGHV500 carrying anti-p21ras scFv [43], and chemically synthesized EpDT3-PEG-Ad5-PTEN (EPAP) [44], in inhibiting the growth of HCC cells. Ad vectors play an important role in the treatment of various diseases including ovarian cancer, glioblastoma, and uterine sarcoma lung cancer when combined with other therapeutic modalities. The use of Enadenotucirev OAd [45] or of ranergene obadenovec (VB-111) [46] combined with paclitaxel for platinum-resistant ovarian cancer treatment has shown a significant increase in median progression-free survival (PFS) and tumor immune cell infiltration. In a mouse model of lung cancer, the delivery of rAd-p53 and IL-2 using an OAd vector, along with concurrent treatment of the chemotherapeutic drug paclitaxel, resulting in significantly inhibited tumor growth [47]. Furthermore, the combination of the rAd-p53 with chemotherapy in the treatment of patients with advanced breast cancer has demonstrated favorable safety and efficacy outcomes [48].

Ad vectors have been successfully utilized in the treatment of patients with breast cancer [49]. In the context of breast cancer mouse models, the utilization of OA AdLyp.sT and mHAdLyp.sT resulted in the safe administration of Ad vector-based gene therapy.

### 2.2. Innate Immune Responses against Adenovirus Vectors

#### 2.2.1. Occurrence of Innate Immune Response

Activation of innate immune responses after Ad vectors infusion has been well documented in clinical trials. For instance, in a clinical trial involving the combined utilization of Ad vectors and chemotherapy for the management of malignant pleural mesothelioma, a substantial 97.5% of patients encountered adverse reactions characterized by cytokine release and interferon syndrome [50]. In a separate clinical trial targeting colon cancer patients with Ad vectors, a surge in the multifunctional cytokine IL-6, signifying the early innate immune response to Ad vectors, was observed within 6 h post administration [51]. Within an hour following intravenous administration of the HAdV vector, the plasma concentration of the complement component C3a, which exhibits heightened efficacy in initiating innate immune signaling, reached its zenith, subsequently provoking systemic thrombocytopenia. Furthermore, upon intravenous delivery of the HAdV vector, the possibility of inducing shock emerged, encompassing symptoms like hypotension, hemoconcentration, tissue edema, and vascular congestion. In a rat model, the virus-triggered lipid mediator platelet-activating factor (PAF) became activated merely 10 min post injection of the HAdV vector, escalating by more than fivefold, eventually culminating in shock within the animals [52].

Ad infection triggers a robust innate immune response, especially during the early stage of viral entry (Figure 2). The innate responses may occur within minutes to hours, resulting in alterations in blood pressure, thrombocytopenia, inflammation, fever, and other related symptoms [3]. Ad entry represents a critical stage in initiating the innate response, as it triggers viral sensing and signaling primarily in the cells of the innate immune system such as macrophages and dendritic cells (DC). Ad vectors can activate numerous innate immune pathways. Toll-like receptors (TLRs) have been identified as key components in the innate sensing of Ad. In plasmacytoid dendritic cells (pDCs), TLR9 is involved in the process of Ad-induced production of IFN, while TLR2, TLR4, and TLR9 induce IL-2 responses when Ad infects mouse or mononuclear phagocyte in vitro [53]. Upon release, type I interferons activate a comparable set of interferon-stimulated genes by binding to IFN-α receptors. The majority of proteins encoded by these genes can regulate signaling pathways or transcription factors, thereby substantially amplifying the synthesis of IFNs, and augmenting the antiviral signal as well as the antiviral state. In essence, the innate immune response provoked by Ad vectors has the potential to diminish the efficacy of gene therapy.

Preexisting antibodies against Ad are prevalent within the population, and individuals who have been infected with the virus develop lifelong immunity, resulting in decreased expression of the transgene and potentially exacerbating virulence of vector transduction. In a study conducted by Maria Bottermann et al. [54], it was discovered that anti-Ad5 antibodies obstructed transgene expression by engaging intracellular Fc receptor tripartite motif-containing protein 21 (TRIM21). Furthermore, the immune response was enhanced by intravenous administration of the vector in individuals with preexisting Ad immunity, and transcriptional analysis revealed that TRIM21 specifically upregulated numerous immune genes, thereby inducing an innate response.

Activation of the innate immune response is independent of the transduction process, and Ad vectors activate the innate immune response in a manner similar to that of biologically active particles. The activation of innate immune response is associated with the capsid proteins on the surface of Ad vectors, and the activation of complement, rather than the genes expressed by the virus. Since most of the viral genome of HAdV is eliminated, transduced cells stop encoding wild-type Ad gene products, including VA-RNA. Thus, the immunotoxicity of HAdV is greatly attenuated, enabling sustained initiation of transgene expression [11,55,56].

After infection with normal non-cancer cells, the OAd exhibits an inability to replicate. However, upon infecting tumor cells, these viruses undergo successful replication, leading to amplified viral production, subsequent release into the tumor microenvironment, and consequential infection of previously uninfected tumor cells [35]. A preclinical investigation has demonstrated the efficacy of Dlta-24-RGD, a cancer-selective oncolytic adenovirus, in the dissolution of malignant gliomas. Within the context of normal immune responses, the viral infection itself and the subsequent viral lysis of cancer cells prompt the release of damage-associated molecular patterns (DAMPs). These DAMPs are identifiable by PRRs expressed on innate immune system cells, culminating in the activation of type I interferon production [57]. Consequently, within an immunocompetent murine glioma model, the oncolytic adenovirus Delta-24-RGD was employed to assess its impact on the immune microenvironment at the tumor site. Findings revealed an augmented population of NK cells at the tumor site post virus injection, indicating the induction of innate immunity due to virus infection [57]. In the treatment of ovarian cancer, oncolytic adenovirus dl922–947 has demonstrated utility. Notably, carvedilol, a drug with the potential to enhance oncolytic adenovirus activity, has been investigated. In a murine ovarian cancer model, the combination of virus treatment with carvedilol exhibited more substantial adenovirus expression compared to virus treatment alone. This combination treatment also led to enhanced infiltration of macrophages and NK cells into the tumor. Consequently, it was inferred that the augmented anticancer effect following combination therapy was attributed to the induction of innate immunity [58].

#### 2.2.2. Evasion of Innate Immune Response

Ad vectors employ strategies to evade innate immune responses, specifically IFN responses and DNA damage responses.

Innate immune evasion strategies of Ad parental viruses include the selection of low immunogenicity serotypes [5,59] and the up-regulation of MYSM1 during viral infection, inhibiting the activation of innate immune signaling pathways [60]. Considering the double-stranded DNA nature of the Ad genome, infection by DNA virus can induce the upregulation of deubiquitinase MYSM1. MYSM1 has the ability to interact with STING and inhibit the ubiquitination of STING K63, thereby leading to the inhibition of the cGAS-STING signaling pathway [60]. Improving the efficiency of Ad delivery can enhance the therapeutic effect by producing more therapeutic proteins and reducing the amount of virus used, thereby inhibiting the activation of the innate immune response. For example, the use of blocking catheters to block hepatic blood flow before Ad vector infusion has shown the potential to enhance the efficiency of vector delivery [5].

Modification of the viral vector to evade the innate immune response is a commonly employed strategy.

Studies have revealed that genome modification can efficiently suppress the activation of an innate immune response [61]. The development of a helper-dependent Ad vector, incorporating numerous gene modifications, has demonstrated increased transgene expression levels and reduced vector toxicity. However, it may trigger an acute inflammatory reaction. To mitigate this adverse reaction, chemical modification of the viral capsid has been explored. For instance, Yasmine Gabal et al. [62] utilized polyethylene glycol (PEG) conjugation to Ad, along with cell-penetrating peptides on the cell surface, to minimize the interaction between the carrier and the host. Consequently, this strategy reduced the likelihood of recognition by immune cells, thereby attenuating the occurrence of the innate immune response. In the case of liposome-encapsulated Ad vectors, PEGylation can offer additional advantages, including the reduction of cytotoxicity, hemolytic activity, anti-vector immunity, and innate immune response [59].

Given the role of complement system activation in promoting the development of innate immunity, inhibiting the activation of the complement system could potentially suppress innate immunity. In pursuit of this objective, Christopher M Gentile et al. [63] conducted genetic modifications to the hypervariable region of the Ad capsid protein. As a result, the capsid surface of the Ad vector displayed the rH17d sequence, thereby inhibiting the classical complement pathway.

The reduction of the likelihood of innate immune response activation can also be achieved through capsid modification of the Ad vector. Since Ad entry into cells primarily hinges on its binding to the coxsackie–adenovirus receptor (CAR), the presence of CARs on various tissue surfaces is constrained. Capsid modification strategies, such as the genetic incorporation of specific motifs or the chemical conjugation of polymers linked to ligands, could potentially enhance gene transduction efficiency into tissues with lower CAR density, consequently diminishing the probability of innate immune response activation [5].

Notably, Svetlana Atasheva et al. [9] employed capsid mutations in a HAdV-C5 vector, affecting the altered HVR1 region. This modified vector displayed no binding affinity to IgM in human and mouse serum and evaded recognition by Kupffer cells following administration [9]. Another approach involved the substitution of the RGD amino acid in the adenovirus penton protein with a laminin-derived peptide incapable of interacting with macrophage β3 integrin. The resultant mutant virus exhibited attenuated cytokine activation in the spleen subsequent to intravenous administration [64]. In summary, structural modifications to the adenovirus capsid protein can effectively circumvent various stages of innate immune recognition, leading to the development of safer adenovirus vectors.

The sequence in the hypervariable region (HVR) of the HAdV5 hexahedron was reciprocally substituted with the corresponding region derived from HAdV48, yielding novel HAdV vectors. These resultant vectors demonstrated considerable immunogenicity even when confronted with elevated levels of preexisting neutralizing antibodies targeting HAdV5. A novel HAdV vector was further engineered by substituting the relevant structure of HAdV5 with the knob structure derived from HAdV3. This modification substantially enhanced the immunogenicity of both the newly created and synthesized HAdV5 vectors [59].

The deletion of E1A has demonstrated significant significance in the suppression of the immune response. Through the incorporation of the E3 region into the recombinant Ad vector, the capacity to mitigate the maximal innate immune response has been achieved, consequently enabling the sustained expression of Ad genes over an extended period [65].

In light of the presence of preexisting neutralizing antibodies, Peng Lv et al. [66] devised a strategy involving bioengineered membrane nanovesicles (BCMNs). They employed in vitro genetic membrane engineering and CRISPR engineering methods on erythrocyte membranes to maintain the infectivity and replication ability of OAd. Notably, encapsulation of OAd with BCMN resulted in a significant reduction in serum levels of IL-6 and TNF-α, indicating that encapsulation of OAd with BCMN significantly inhibited the OAd vector-induced innate immune response by protecting the OAd surface from the immune system.

## 3. AAV Vector Therapy

### 3.1. Introduction to AAV Vector

#### 3.1.1. Principle of Action of AAV Vector

AAV is composed of a 26 nm diameter icosahedral protein capsid and a single-stranded DNA (ssDNA) genome measuring approximately 4.7 kb in length (Figure 1). At each end of the genome, there are inverted terminal repeat (ITR) sequences spanning 145 nucleotides. These ITR sequences adopt a hairpin structure, which is a cis-acting element essential for initiating DNA replication and packaging the recombinant AAV genome into infectious virions. Positioned between the ITR sequence is the viral coding region, which encompasses two open reading frames (ORFs). These ORFs give rise to the replication protein Rep, capsid protein Cap, and the packaging activation protein AAP [67]. The Rep gene encodes four non-structural proteins: Rep78, Rep68, Rep52, and Rep40, which are mainly involved in viral genome replication and integration into the host genome. The Cap gene, on the other hand, encodes three structural capsid proteins, VP1, VP2, and VP3. Assembly activation protein AAP primarily facilitates viral genome packaging and the subsequent release from host cells [68]. Among the structural capsid proteins, VP1, VP2, and VP3, there exist 60 copies in a ratio of VP1: VP2: VP3 = 1:1:10. AAV is incapable of self-replication and relies on co-infection with other helper viruses for replication, such as Ad and herpes simplex virus (HSV), for its replication process.

Based on variations of AAV capsid protein, AAV can be categorized into 13 wild types and over 100 mutants [69]. Each serotype exhibits distinct infection efficiencies across various tissues and cell types. Notably, AAV demonstrates exceptional stability, rendering it more resistant to temperature and pH fluctuations compared to other viral vectors. Furthermore, AAV exhibits lower immunogenicity than other viruses [70], and most AAV vectors do not integrate their DNA into the genome of patients. Consequently, the risk of insertion mutations associated with AAV vectors is minimal. These inherent properties render AAV highly suitable for specific gene therapy applications.

The primary AAV utilized in gene therapy is recombinant AAV (rAAV), which has the same capsid sequence and structural organization as wild-type (WT) AAV. However, the genome packaged within rAAV has undergone modifications, wherein the AAV protein coding sequence is deleted and a therapeutic gene expression cassette is inserted. Notably, ITR represents the only viral-derived sequence present in rAAV. Furthermore, rAAV exhibits optimal genome loading capacity, accommodating sequences of up to 5.0 kb or less [2].

Although AAV is less immunogenic than Ad, the capsid proteins, as well as the delivered nucleic acid sequences, can trigger various components of our immune systems. Most people have already been exposed to AAV and have already developed an immune response to the specific variant to which they were previously exposed, resulting in a pre-existing immune response in the sense that neutralizing antibodies (NAbs) against the AAV are already present. NAbs against AAV2 are the most prevalent in the human body, and the prevalence of NAbs against AAV8 and AAV5 is the lowest, so these serotypes are more conducive to use as gene therapy vectors [71].

#### 3.1.2. Application of AAV Vectors in Gene Therapy

A range of approved AAV vector-based drugs is presently available (Table 1). In 2012, Glybera marked the first recommendation of an AAV vector gene therapy for treating lipoprotein lipase deficiency [17]. Nonetheless, due to economic considerations, Glybera was withdrawn from the market in 2017 [72]. In 2017, Luxturna, an AAV2 vector gene therapy, received FDA approval for addressing Leber congenital amaurosis attributed to RPE65 mutations [18,73]. In 2019, Zolgensma, a scAAV-SMN1 gene therapy, gained FDA approval to treat spinal muscular atrophy (SMA) [20]. Eladocagene exuparvovec (Upstaza), a gene therapy drug vector founded on rAAV2-hAADC, obtained EMA approval in 2022 to address severe ADCC deficiency in patients [24]. Likewise, in 2022, valoctocogene roxaparvovec (Roctavian), a gene therapy agent vector-built upon AAV5-hFVIII-SQ, secured EMA approval for treating severe hemophilia A (congenital factor VIII deficiency) in adults without a history of FVIII inhibitors and without detectable antibodies to AAV5 [25].

Most of the current studies involve the direct administration of AAV vectors to patients [74]. Gene therapy has emerged as a promising therapeutic approach for hemophilia, and AAV2 stands as the pioneering vector employed in its treatment [75]. Furthermore, AAV3 and AAV5 exhibit potential for hemophilia therapy, demonstrating improvements in disease severity [76,77,78]. AAV8 is considered the optimal vector for the treatment of hemophilia [79]. Compared with AAV2 and AAV5, AAV8 boasts a lower prevalence in human seroprevalence, elicits a reduced immune response towards the vector capsid, and exhibits a pronounced affinity for the liver. Consequently, AAV8 facilitates efficient transduction of hepatocytes when administered to animal models through the peripheral circulation [80,81].

AAV holds promise in the treatment of ocular disorders. Given the highly compartmentalized nature of the eye and the absence of lymphatic vessels, it is generally believed that the risk of activating the innate immune response is low when AAV vectors are administered into the eyes [82]. It has been reported that subretinal vector delivery is associated with lower immunogenicity compared to intravitreal injection [83]. Studies employing subretinal injection of AAV2 and AAV8 in patients with Leber’s congenital amaurosis, retinal degeneration, and CNGA3-linked achromatopsia have demonstrated improvements in visual acuity and favorable safety profiles [84,85,86].

AAV vectors have emerged as a potential therapeutic option for neurological disorders. Commonly utilized vectors include AAV1, AAV2, AAV5, and AAV8. AAV2, as a first-generation vector, has demonstrated its safety and efficacy in the treatment of various neurological conditions, including Parkinson’s disease [87], Alzheimer’s disease [88,89], AADC deficiency [90], and several genetic diseases. As the second-generation vector, AAV5 has shown promise in ameliorating Huntington’s disease (HD) [91] and spinocerebellar ataxia type 3 (SCA3), respectively [92]. AAV9, belonging to the third generation of vectors, exhibits the capability to efficiently target the brain and spinal cord. Compared with other serotypes, AAV9 can demonstrate enhanced blood–brain barrier penetration following intravenous administration, offering the potential for minimally invasive therapeutic interventions [93].

AAV vectors have emerged as a promising approach for the treatment of muscular diseases. For instance, AAV9 has shown efficacy in reducing the severity of spinal muscular atrophy (SMA) in patients [94]. In the case of Duchenne muscular dystrophy (DMD), patients treated with rAAVrh74 experienced significant improvements in their motor abilities after one year of treatment [95]. Furthermore, the administration of AAV8 vectors to patients with X-linked myotubular myopathy (XLMTM) resulted in the recovery of exercise capacity [96].

AAV vectors have also shown potential in the treatment of hearing impairment. Studies utilizing AAV1-VEGFA165, exo-AAV1-GFP, AAV9-PHP.B, and other vectors have demonstrated improved blood supply and alleviation of hearing loss in animal models, highlighting their safety and therapeutic potential in this field [97,98,99].

### 3.2. Innate Immune Responses against AAV Vectors

#### 3.2.1. Occurrence of Innate Immune Response

Despite the diminished innate immune response observed in AAV compared to Ad vectors, the host immune response remains a significant hurdle in achieving sustained and effective therapeutic gene expression [100]. These immune responses encompass cytotoxic T lymphocyte (CTL) responses against AAV capsids and therapeutic proteins, the presence of NAbs against AAV viral particles, the production of antibodies against therapeutic proteins, and innate immune responses triggered by AAV transduction [101].

Observation of activated innate immune responses subsequent to AAV vector infusion has also been documented in clinical trials. In a clinical trial involving AAV5-hFIX gene therapy among adults with hemophilia B, the onset of fever and elevated ALT levels within 24 h post-vector delivery indicated the activation of the innate immune response [77].

Previous investigations demonstrated that administration of a high dosage of AAV vector intravenously to mice for 1 h elicited escalated transcription of genes associated with inflammatory cytokines and chemokines, including TNF-α, RANTES, MIP-1β, MIP-2, MCP-1, and IP-10 [102]. 

TLR9-mediated sensing of CpG motifs in the AAV genome is likely to occur during endosomal trafficking, where partially exposed virions can be recognized (Figure 3). Alternatively, viral capsid degradation within lysosomes can expose the genome to TLR9, leading to the activation of nuclear factor kappa-B (NF-κB) and interferon regulatory factor 7 (IRF7) via myeloid differentiation primary response protein88 (MyD88) signaling pathway. This activation ultimately regulates the production of IFN-I and ISGs [103]. In addition to TLR9 activation, an upregulation in the transcription of TLR2, which recognizes microbial proteins and glycolipid structures, has also been observed following rAAV infection, suggesting the potential involvement of TLR2 in innate immunity against rAAV vectors [13,104].

The presence of NAbs not only reduces the efficiency of vector transduction and compromises the therapeutic effect but also triggers complement activation in the presence of NAbs against the capsid. The complement system, consisting of a variety of proteins, constitutes a significant component of the host’s innate immune system, encompassing the classical pathway and the lectin pathway. While it is commonly believed that the complement system plays a limited role in the innate immune response against AAV, Manish Muhuri [13] argues that the evidence from pathological observations suggests the activation of complement in the human body in response to AAV.

The utilization of self-complementary AAV (scAAV) vectors enhances the efficacy of the vector by facilitating faster and more robust transgene expression. Thus, in turn, allows for the administration of lower and safer vector doses. However, when compared to the single-stranded AAV (ssAAV) vector, the use of the scAAV genome leads to an augmented innate immune response towards the transgene, primarily due to activation of the TLR9/MyD88 pathway. The DNA component of the AAV vector, as well as the potential production of dsRNA [106], resulting from the ITR promoter activity, can function as an adjuvant, activating innate immunity alongside other host-specific factors. The buildup of dsRNA would subsequently trigger the MDA5 sensor within human hepatocytes transduced with AAV, resulting in the induction of type I interferons (IFNs) expression [83,107]. Following their release, type I IFNs activate a similar array of ISGs through their interaction with IFN-α receptors. A significant portion of the proteins encoded by these genes function in the regulation of signaling pathways or transcription factors, thereby substantially amplifying IFN synthesis and augmenting both the antiviral signal and antiviral state. In summary, the innate immune response elicited by AAV vectors may lead to decreased efficacy of gene therapy.

The ITR structure of the AAV genome may activate cytoplasmic DNA receptors. Studies have demonstrated that rAAV can induce cGAS and antiviral genes such as TNF-α and IFN-γ [105]. The ITR, located at either end of the AAV or within the bidirectional promoter, can generate reverse complementary positive-strand RNA and negative-strand RNA. These RNA molecules form dsRNA in target cells, subsequently triggering the innate immune response mediated by the dsRNA recognition receptor. As cytoplasmic RNA sensors, RLRs can activate downstream innate immune signaling pathways through adaptor protein MAVS. RLRs include three RNA-binding proteins: MDA5, RIG-I, and LGP2. MDA5 and RIG-I can recognize the dsRNA formed by AAV, thereby activating the RLR-MAVS signaling pathway to induce the production of IFN-β [13].

#### 3.2.2. Evasion of Innate Immune Response

Various strategies exist that do not entail genetic alteration of AAV vectors but aid in circumventing the innate immune response. For instance, within AAV gene therapy, distinct serotypes exhibit varying affinities for different tissues, thereby necessitating the choice of serotypes compatible with specific disease contexts [5]. Additionally, the adoption of less immunogenic vector delivery approaches can be considered [83]. 

Given that AAV serotypes exhibit differential tissue tropism, selecting an appropriate serotype in line with the target tissue becomes imperative in gene therapy. For instance, employing AAV8 vectors for treating liver disorders can enhance the efficacy of gene therapy [5]. The immunogenicity of AAV vectors is partly dependent on the dosage administered. Low doses of vectors are more likely to induce mild inflammation, which can be controlled and dose not lead to complete loss of transgene expression [83]. The transgene immune response, which plays a crucial role in the production of anti-capsid antibodies, is influenced by various factors, including the AAV capsid, the method of vector delivery, and the tissue specificity of the promoter driving gene expression. Notably, systemic, and intramuscular vector delivery methods have shown greater immunogenicity compared to systemic delivery with gene transfer to immune organs or the use of liver-specific promoters [83].

Utilizing immunosuppressive agents in gene therapy can impede DNA synthesis and cellular signaling essential for lymphocyte activation and proliferation. For instance, employing glucocorticoid dexamethasone can mitigate the innate immune response, thereby curbing cytokine storms [108].

Furthermore, altering the viral structure to evade the innate immune response stands as a commonly employed approach.

In clinical trials involving AAV vectors, the used of corticosteroids has been a common approach to suppress the immune response [109]. Alternatively, more targeted approaches have been explored, such as modifying the capsids for immune evasion through techniques like site-directed mutagenesis, directed evolution, and utilizing exosomes. To reduce the presentation of AAV2 capsid-derived peptide epitopes on MHC class I, the proteasome inhibitor bortezomib has been employed in hepatocytes [109].

Inhibiting TLR9 activation can diminish innate immune activation probability. In a study, a short single-stranded DNA oligonucleotide (TLR9i) antagonizing TLR9 activation was employed. TLR9i sequence was inserted into scAAV vector plasmid encoding human Factor IX (FIX) to form scAAV8.FIX.io1. In mice, equal amounts of scAAV8.FIX and scAAV8.FIX.io1 showed minimal innate immune response activation in the liver. Compared to the PBS injection group, scAAV8.FIX.io1 injection did not elevate IFN gene expression after 4 h. The TLR9i-modified single-stranded AAV2 vector (AAV2.GFP.wpre.io2) was used with primary human peripheral blood mononuclear cells from 13 healthy individuals. AAV2.GFP.wpre induced a stronger cytokine response than TLR9i modification [100]. Recent accounts indicate that the integration of TLR9 inhibitory sequences has led to reduced immune reactions related to rAAV in mice and pigs [13].

Narendra Maheshri et al. conducted AAV capsid engineering to generate AAV2 variants with enhanced attributes. Their findings highlighted that these AAV mutants exhibited altered affinities for heparin and demonstrated heightened efficacy in evading neutralizing antibodies to AAV. This manipulation also bolstered gene delivery, even in the presence of preexisting neutralizing antibodies to AAV within the population [110].

Research has evidenced that restraining the recognition signaling of TLR9 through ODN antagonists and inhibiting the NF-κB pathway can decrease innate immune responses, although full elimination remains elusive. Predominantly, NF-κB inhibitors impede the initial TLR9 activation while not fundamentally curtailing IFN production. In efforts to mitigate TLR9-mediated innate immune responses, AAV transgene cassettes are tailored to diminish TLR9 recognition of CpG sites. This strategic approach has been applied in clinical trials involving patients with hemophilia [111].

## 4. Lentiviral Vector Therapy

### 4.1. Introduction to Lentiviral Vectors

#### 4.1.1. Principle of Action of Lentiviral Vector

LV vectors are spherical enveloped viruses, with a diameter ranging from 80 to 120 nm. They carry a single-stranded RNA genome with two copies. LV belongs to the retrovirus genus, capable of infecting both dividing and non-dividing cells, and exhibits a relatively prolonged incubation period and low pathogenicity [112]. Reverse transcriptase encoded by the LV genome transcribes viral RNA into dsDNA, which is subsequently integrated into the host cell genome by viral integrase [112]. The HIV genome consists of two positive-sense RNA strands, approximately 9 kb in length, encoding nine regulatory and auxiliary genes gag, pol, env, rev, tat, vpr, vpu, vif, and nef. Within the HIV genome, trans elements encode functional, structural, and accessory proteins, while non-coding cis elements such as the long terminal repeat (LTR) are also present.

LV vectors can be categorized into primate LV vectors and non-primate LV vectors based on the source of the virus. Primate LV vectors consist of human immunodeficiency virus type 1 (HIV-1), type 2 (HIV-2), and simian immunodeficiency virus (SIV). Among these, HIV-1 is one of the most widely studied viruses. LV vectors share common characteristics in terms of viral structure, phagocytosis, and the elicited immune response [113].

In the current third generation of LV vectors, a conditional packaging platform is created by removing the tat and a portion of the 3′-LTR, while providing the rev gene required for replication in trans. Furthermore, the U3 region in the 5′-LTR of the transfer plasmid is replaced with a constitutive promoter, enabling transcription in the absence of tat, thereby improving safety and reducing immunogenicity [114,115,116].

LV generally primarily targets immune cells, particularly macrophages and T lymphocytes [113]. HIV-1 mainly infects and replicates within CD4^+^ T cells, with a smaller fraction infecting CD4^+^ DCs and macrophages.

#### 4.1.2. Application of Lentiviral Vectors in Gene Therapy

A range of approved lentiviral (LV) vector drugs currently exist. In 2017, the FDA granted approval to Kymriah (tisagenlecleucel), a gene therapy drug rooted in LV-CD19, intended for managing acute lymphoblastic leukemia and diffuse large B-cell lymphoma [19]. The year 2020 saw the EU’s endorsement of Libmeldy, a gene therapy drug grounded in SIN LV, for addressing metachromatic leukodystrophy (MLD) [21]. Similarly, in 2021, the European Union approved Elivaldogene autotemcel (Skysona, eli-cel), a gene therapy drug built on LV. ABCD1 is designed for managing cerebral adrenoleukodystrophy (CALD) [22]. Breyanzi (Lisocabtagene Maraleucel), a gene therapy drug founded on LV-CD19, was also sanctioned by the FDA in 2021, targeting patients with relapsed or refractory large B-cell lymphomas [23]. Additionally, in 2021, the FDA endorsed Abecma (Idecabtagene Vicleucel, ide-cel), a gene therapy drug employing LV-CD19 as a vector, for treating relapsed or refractory multiple myeloma (R/R MM) [23]. Finally, in 2022, Zynteglo (Betibeglogene Autotemcel, beti-cel), a gene therapy drug hinging on the BB305 LV vector, received FDA approval for managing β-Thalassemia patients [27].

LV vectors have been utilized as gene therapy vectors in the treatment of genetic disorders. It has been applied to the treatment of chronic granulomatous disease (CGD). In a study by Donald B Kohn et al. [117], LV vectors were first utilized to treat X-linked CGD (X-CGD), resulting in the majority of patients being able to discontinue antibiotic prophylaxis without associated infections. For patients with transfusion-dependent β-thalassemia (TDT), LV vector gene therapy has proven to be a safe and effective treatment. Autologous CD34^+^ cells transduced with LV vector BB305 have been administrated to patients with TDT [118,119]. The findings demonstrate that a substantial number of patients achieve remission of clinical response, experienced a reduction in transfusion frequency, or even discontinued transfusions altogether. The treatment of infants with server combined immune-deficiency (SCID) through LV vector transduction has demonstrated remarkable clinical success. The administration of LV-transduced CD34^+^ cells to infants with SCID, followed by targeted low-exposure busulfan infusion, effectively restored immune function and facilitated normal growth with minimal toxicity and high safety level [120,121]. Mutations in ATP binding cassette subfamily D member 1 (ABCD1) lead to the loss of adrenoleukodystrophy (ALD) protein function, resulting in X-linked adrenoleukodystrophy, Florian Eichler et al. [122] transduced autologous CD34^+^ cells from patients with LV vectors and observed the expression of ALD protein in all patients, with majority patients surviving and experiencing few adverse reactions. LV vector gene therapy may serve as a viable alternative to allogeneic hematopoietic stem cell transplantation for this patient population.

Allogeneic hematopoietic stem cell transplantation is also considered the standard treatment for Hurler syndrome, known as mucopolysaccharidosis type I Hurler variant (MPSIH). However, this treatment only provides a partial cure and is associated with a higher incidence of adverse reactions. In a study conducted by Bernhard Gentner et al. [123], autologous hematopoietic stem and progenitor cells were transduced with LV vectors encoding α-L-iduronidase (IDUA) in eight children with MPSIH. Interim findings indicated that the safety profile of this approach was comparable to that of allogeneic hematopoietic stem cell transplantation. Moreover, gene therapy resulted in previously undetectable levels of IDUA activity and demonstrated improvements in the patients’ motor abilities and cognitive performance. In another application case, the use of LV-transduced bone marrow-derived CD34^+^ hematopoietic stem cells (HSCs) in the treatment of children with leukodystrophy showed promising results. This therapeutic approach effectively decelerated disease progression, ameliorated neurological abnormalities, and proved to be safe, delivering substantial treatment benefits [124].

LVs have been investigated in clinical trials as vectors for cancer immunotherapy. In these trials, LVs are used to modify patients’ immune cells, such as T cells, to express chimeric antigen receptors (CARs) or T-cell receptors (TCRs) targeting specific tumor antigens. Acute lymphoblastic leukemia (ALL) carries a bleak prognosis upon relapse following allogeneic transplantation. In an effort to address this, LV vectors were employed to transduce a CAR encoding CD3ζ and 4–1BB. Subsequently, gene therapy using LV vectors yielded a marked enhancement in the long-term survival of patients [125]. LV vector-transduced fully human belantamab mafodotin (BCMA)-specific CART cells were utilized to treat multiple myeloma (MM) clinically. The results demonstrated that all patients successfully expanded CART-BCMA cells [126].

Preclinical investigations have explored the application of LV vector gene therapy for several diseases prior to clinical implementation. Promising outcomes have been observed in murine models of breast cancer [127], liver cancer [128], bladder cancer [129], hemophilia [130,131,132,133], and AD [134,135,136], following treatment with LV vectors.

### 4.2. Innate Immune Responses against Lentiviral Vectors

#### 4.2.1. Occurrence of Innate Immune Response

LV vectors are mainly derived from HIV-1, necessitating a comprehensive examination of the innate immune response following HIV-1 infection (Figure 4).

The initiation of innate immune responses following the administration of lentiviral (LV) vectors has been documented in clinical trials. For instance, in a clinical trial involving a patient with relapsed and refractory acute myeloid leukemia who underwent LV vector treatment, pronounced fever episodes manifested within 0.5 to 1 h following daily vector infusion, indicating the activation of innate immune responses triggered by LV vector gene therapy [137]. In a preclinical experiment involving mice, intravenous injection of LV vector induced a swift and transient interferon (IFN) response, marked by the production of IFN-α and IFN-β [138]. Within four hours of intravenous administration of LV vector to mice, a rapid and transient upsurge of type I interferon was detected in their serum [102].

PRRs in host cells play a crucial role in recognizing PAMP to drive IFN-mediated immune responses against the virus. Multiple PAMPs associated with HIV-1 have been identified, including capsids, genomic RNA, reverse transcription products such as ssDNA, dsRNA, and RNA-DNA hybrids [139]. When the LV vectors reach the target nucleus and integrate into the host genome, innate sensors recognize LV nucleic acids and proteins. Notably, the transduction of LV vectors into immune cells and stem cells activates innate immune responses through TLR recognition [140]. Certain sensors such as TLR7, TLR8 and unidentified DNA sensors generally recognize nucleic acids associated with retroviral infection. 

**Figure 4 viruses-15-01801-f004:**
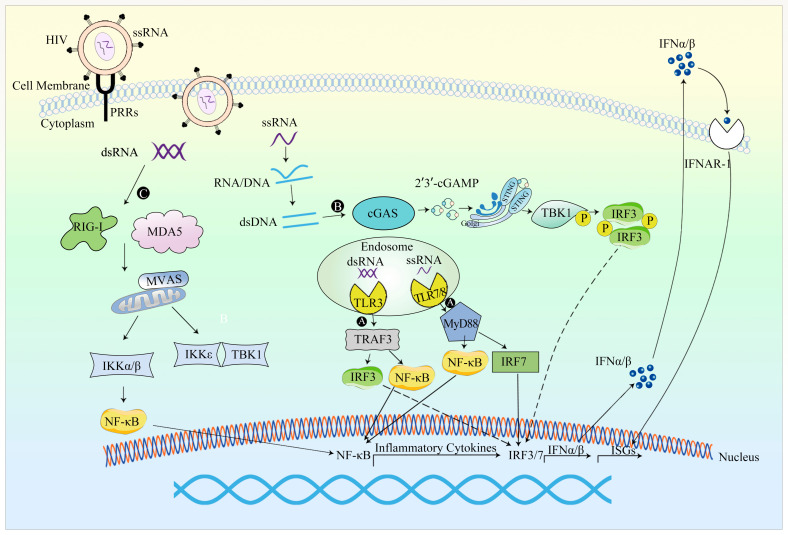
An overview of the innate response to LV vectors. (A) LV RNA genome can activate TLR3, TLR7 and TLR8, resulting in production of IFN-I and activating innate immunity [140]. (B) In certain cell types, reverse transcription gives rise to cytoplasmic HIV DNA, thereby leading to the activation of the cGAS-STING innate immune signaling pathway [139]. (C) The secondary structure of HIV-1 genome RNA can elicit innate immune responses similar to those triggered by full-length genome RNA, activating the RNA sensor RIG-I in primary human peripheral blood mononuclear cells and macrophages. The secondary structure of HIV-1 genome RNA can activate the RNA sensor RIG-I in primary human peripheral blood mononuclear cells and macrophages. Thus, RIG-I-MAVS innate immune signaling pathway is activated [139,141].

In humans, preexisting immunity to lentiviral (LV) vectors is generally limited. However, upon the transduction of LV vectors, the nucleic acids and proteins within the vector can be identified by restriction factors (RFs) which serve as innate immune sensors [142].

HIV-1 RNA, encompassing both genome RNA and newly synthesized RNA, has the capacity to engage RNA sensors. The secondary structure of HIV-1 genome RNA can elicit innate immune responses similar to those triggered by full-length genome RNA, activating the RNA sensor RIG-I in primary human peripheral blood mononuclear cells and macrophages. Thus, the RIG-I-MAVS innate immune signaling pathway is activated [139,141]. Notably, Andrea Annoni et al. [143] demonstrated that the LV RNA genome can activate TLR3 and TLR7, resulting in the production of IFN-I and activating innate immunity.

In certain cell types, reverse transcription gives rise to cytoplasmic HIV DNA. Acting as a PRR, cGAS selectively binds to the stem-loop structure of HIV-1 ssDNA in a sequence-specific manner and becomes activated by the reverse transcription product, thereby leading to the activation of the cGAS-STING innate immune signaling pathway [139].

LATS1/2, the core kinase of the Hippo pathway, has been implicated in the regulation of anti-tumor immunity. Tiansheng He et al. [144] discovered that LATS2 has the ability to interact with PQBP1, a cofactor of cGAS. This interaction leads to the augmentation of the antiviral response mediated by the cGAS-STING signaling pathway, subsequently resulting in enhanced IFN-I production. Consequently, LATS2 plays a pivotal role in promoting the development of innate immune responses.

The magnitude of the innate immune response can be further augmented by additional factors. These factors encompass an elevated vector dose exceeding the initial dosage administered during HIV-1 infection, loss of viral accessory proteins that aid in immune evasion during HIV-1 infection, and the stimulation of the cGAS pathway through the presence of DNA within the vector [140].

#### 4.2.2. Evasion of the Innate Immune Response

The evasion of innate immunity in parental viruses is exemplified by the m6A modification of the HIV virus, which inhibits the activation of IRF3 and IRF7 [145]. N6-methyladenosine (m6A) is a type of RNA modification that plays a significant role in various biological processes. HIV-1 RNA, which contains m6A modifications, has been observed to regulate the viral infection of CD4^+^ T cells. A study conducted by Shuliang Chen et al. [145] investigated the impact of m6A modification of HIV-1 RNA. It was found that these modifications suppressed the activation levels of the upstream transcription factors IRF3 and IRF7, which are involved in the production of IFN-I and are crucial for evading RIG-I-mediated RNA sensing. Consequently, this evasion strategy facilitated the escape from innate immune responses.

Similarly, akin to AAV and Ad, the evasion of innate immune responses can be attained through the modification of LV vector or drug regulation.

LVs produced by MHC-free 293T cells or CD47hi LVs were administered intravenously to rhesus monkeys. The results revealed that CD47hi LV-treated non-human primates exhibited reduced elevation of macrophage-associated cytokines, indicating a safer and well-tolerated response. This result suggests that CD47hi LV has the potential to mitigate the activation of the innate immune system [146].

Furthermore, studies have demonstrated that LV vectors modified by the CD47ED fusion gene, incorporating extracellular domain-core of CD47 and streptavidin, can trigger phagocytosis of LV by antiphagocytic cells [147]. Notably, when BX795, an inhibitor of RIG-I, MDA5, and TBK1/IKKε complex downstream of TLR3, was employed to inhibit innate immune signals in cell lines, it resulted in significantly improved LV gene modification efficiency and enhanced LV transduction efficiency. This practical and safe approach shows promise for future applications [148].

At present, the most commonly used method for transduction of human hematopoietic stem and progenitor cells (HSPCs) involves the use of LVs. However, it has been observed that the expression of LVs in stem cells can trigger the innate immune response. Carolina Petrillo et al. [149] found that cyclosporine H (CSH), a new drug, could enhance the efficiency of LV transduction and gene editing in HSPCs. Notably, CSH effectively blocked the IFN response induced by LV vectors, demonstrating a favorable anti-innate immune response effect.

In a preclinical investigation, Judith Agudo et al. found that mice receiving treatment with GC dexamethasone (Dex) concurrent with LV vector administration exhibited significant upregulation of IFN-induced genes in LV-treated mice. Notably, in the liver of LV + Dex-treated mice, the expression of these genes was normalized. Furthermore, the total number of neutrophils and macrophages in the liver was decreased, indicating that the administration of GC could effectively impede the innate immune response triggered by the LV vector [150].

In order to reduce the innate immune response caused by LV vectors, specific antibodies can be used to block type I interferon receptor, IL-6, IL-6 receptor, and other specific antibodies, and applied to LV vectors gene therapy can reduce the occurrence probability of innate immune response to LV vectors [143].

## 5. Conclusions and Prospects

After decades of research and development, the application of viral vectors, including Ad, AAV, and LV vectors, in gene therapy has made significant strides and is now extensively employed in clinical settings. However, the innate immune response to those vectors and transgenes poses challenges to improving therapeutic efficacies. Investigating the underlying mechanisms of these innate immune responses can pave the way for vector evasion, thereby expanding the scope of gene therapy to encompass a wider range of diseases. At present, ongoing research has demonstrated that the utilization of inhibitors or modification of viral vectors can facilitate vector evasion of the innate immune response, ultimately enhancing the effectiveness of gene therapy. Nonetheless, several factors hinder the optimal efficacy of gene therapy. Notably, preexisting vector immunity and the propensity for augmented innate immunity at elevated vector dosages emerge as substantial limitations within clinical applications, exerting potential impact on the gene therapy vectors’ effectiveness. In the realm of gene therapy, the comprehension of host- and vector-related determinants that intricately influence the genesis of cytotoxic reactions, thereby contributing to the instability of transgene expression duration, remains limited [107]. Endeavors aimed at refining viral vector transduction strategies, particularly towards the utilization of diminished dosages, persist as requisite measures to enhance therapeutic efficacy, a notion especially relevant in clinical settings [142]. Hence, further investigations are imperative, encompassing not solely the scrutiny of the innate immune response, but also a comprehensive exploration of the adaptive immune response. Such in-depth inquiry holds the potential to ameliorate the therapeutic impact of gene therapy.

## Figures and Tables

**Figure 1 viruses-15-01801-f001:**
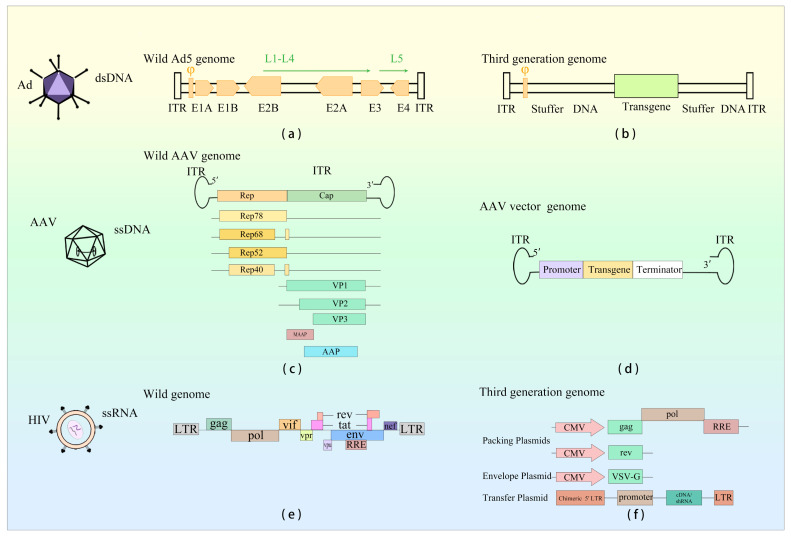
Genome structure description of Ad, AAV, and LV viruses and their viral vectors. (**a**) Adenovirus genome composition. (**b**) Genome composition of third-generation adenovirus vectors currently used for gene therapy. (**c**) Adeno-associated virus genome composition. (**d**) Genome composition of recombinant AAV vectors currently used in gene therapy. (**e**) Genome composition of HIV. (**f**) Genome composition of third-generation lentiviral vectors presently utilized in gene therapy.

**Figure 2 viruses-15-01801-f002:**
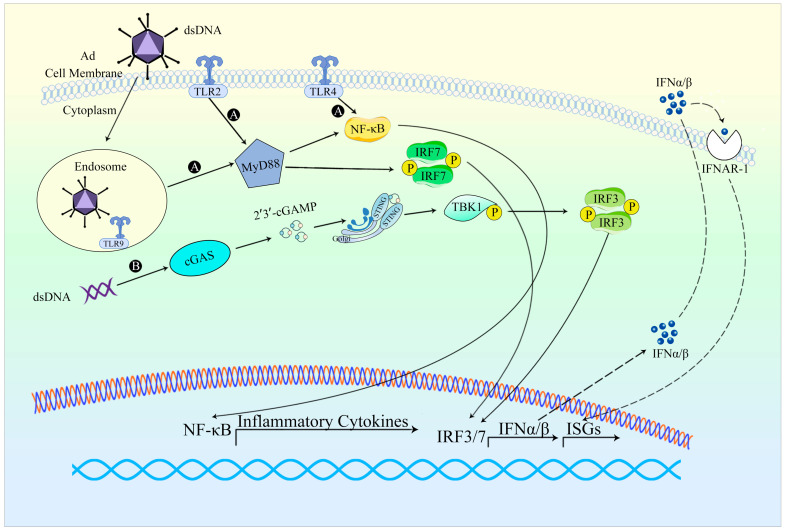
An overview of the innate response to Ad vectors. (A) Upon Ad infection, the genomic DNA of Ad can serve as a PAMP, leading to the activation of TLR2 and TLR9. TLRs subsequently trigger the activation of NF-κB and IRF7. This cascade of activation culminates in the regulation of the production of IFN-I and ISGs [53]. (B) DsDNA within the Ad genome propels the engagement of the cGAS-STING signaling pathway [10].

**Figure 3 viruses-15-01801-f003:**
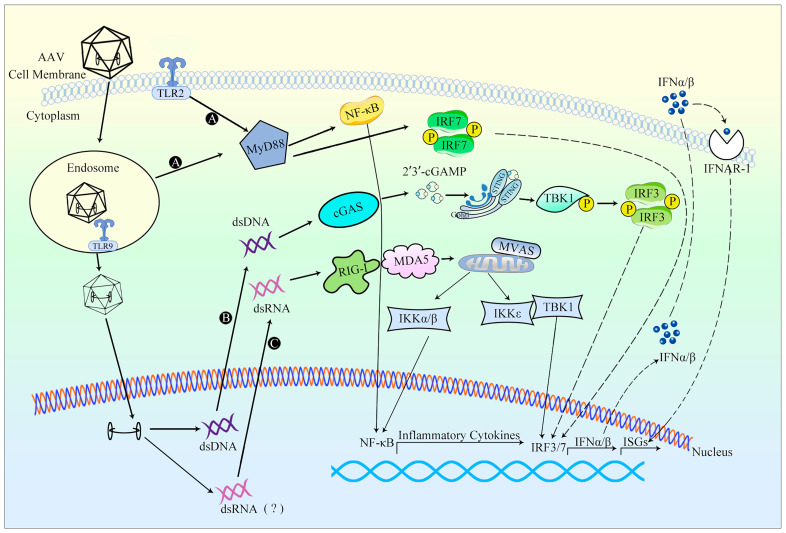
An overview of the innate response to AAV vectors. (A) Upon AAV infection, AAV’s genome DNA can act as PAMP, leading to the activation of TLR2 and TLR9. Then, TLRs activate NF-κB and IRF7 via MyD88 signaling pathway, this activation ultimately regulates the production of IFN-I and ISGs [13,103,104]. (B) The ITR structure of the AAV genome may activate cytoplasmic DNA to induce cGAS and antiviral genes and activate the cGAS-STING signaling pathway [105]. (C) Due to the promoter activity of the ITR of AAV, AAV may form dsRNA in target cells and trigger the RIG-I/MDA5-mediated RLR-MAVS innate immune signaling pathway [83,106]. The “(?)” in the figure implies that dsRNA could be theoretically formed during this process, but has not been demonstrated so far.

## Data Availability

Not applicable.

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
