# Peer review of "Innate Immune Response to Viral Vectors in Gene Therapy"

_viruses, 2023, doi:10.3390/v15091801_

Round 1
Reviewer 2 Report
In the review “Innate immune Response to viral vectors in Gene Therapy”, Y. Wang and W. Shao describe 3 viral vectors derived from adenovirus, AAV and lentivirus that are mainly used for gene therapy and the associated innate immune system activation.
Overall, the topic is interesting and the authors give an overview of the different innate sensors or signaling pathways involved in the innate immune response. Citing specific adverse events that occurred in clinics will strengthen this manuscript.
Major points:
1- A table summarizing the authorized drugs using the 3 cited viral vector must be provided to demonstrate these vectors are efficient. These drugs and their application must be cited in the introduction of each “application of X vectors in gene therapy” sections.
2- The review focuses on innate immune responses to viral vectors that are mainly described in patients but the authors don’t give an overview of the literature regarding these aspects. They must at least give one example for each viral vector.
3- The conclusion of the manuscript suggests that the ongoing research allows immune evasion. The authors must moderate their message since there are still unresolved immune issues to overcome.
Minor points:
1- A cartoon with the viral genome structures for each virus will be helpful.
2- A cartoon summarizing the signaling pathways involved after adenovirus recognition is missing.
Reviewer 3 Report
This review by Wang and Shao is focused on a very interesting topic, i.e. the innate immune response against viral vectors exploited in gene therapy approaches. Although there are already some reviews in the literature dealing with the same issue (e.g. Shirley JL, de Jong YP, Terhorst C, Herzog RW. Immune Responses to Viral Gene Therapy Vectors. Mol Ther. 2020 Mar 4;28(3):709-722. doi: 10.1016/j.ymthe.2020.01.001. Epub 2020 Jan 10. PMID: 31968213; PMCID: PMC7054714.) this manuscript could still be original and give further insights to the field. However, to achieve this aim, the Authors should address the following main points:
1. the title implies a focus on "innate immunity" that is quite interesting. However, in the manuscript the Authors are mixing up inflammatory response, acquired and innate immunity. This is quite confusing and lowers the quality and originality of the review. I would suggest to focus on innate immunity introducing a paragraph clearly explaining what is innate immunity against viruses, and in particular against the viral families that will be taken into account in the following paragraphs.
2. The Authors need to better separate the innate response against viral vectors from the one triggered by the parental virus. Vectors usually lack several genes/proteins typical of the parental virus and this has an impact also on the mechanisms of innate immunity put in place by the host cells. As it reads now, the manuscript is not clear enough on this aspect
3. If the Authors want to discuss also oncolytic adenoviruses, they have to clearly describe their main differences with the adenoviral vectors, as once again, these influence the mechanisms of innate response
4. one of the most interesting and original aspect of the review would be a description and discussion of the approaches developed to overcome the innate response against viral vectors. However, in the current version of the manuscript, this part is too limited. Furthermore, once again, it is not always easy to distinguish between mechanisms evolved by the parental virus to overcome innate immunity and modifications made to the viral vectors to achieve a better escape.
5. A better discussed of whether and how the innate immune response has an impact on gene therapy protocols failure is required.
6. more Figures and Table are necessary to make the take-home messages clearer and easier to get
In conclusion, before this review could be suitable for publication in Viruses, I would suggest to the Authors an extensive revision to stress the novelty of their work. For instance, they could start with a paragraph describing innate immunity against viruses with a focus on the 3 viral families taken into consideration. At this point they could brifly describe ech relevant parental virus. They could discuss here the mechanisms evolved by each partental virus to escape innate immunity. Here 1/2 Figure/s summarizing the main mechanisms described and/or 1/2 Tables should be added. Next the Authors can introduce each vector, the innate immunity against it along with its impact on the success of gene therapy protocols/adverse events and the relevant modifications that have been put in place to overcome this host response. For each vector, 1 Figure or 1 Table summarizing the main messages present in the text should be introduced.
A moderate editing of English language is required to make some sentences easier to understand. Please check the text carefully for spelling mistakes.
Round 2
Reviewer 1 Report
Dear authors,
The authors tried to reflect the reviewer's comments in the very good way. But there are still some problems and mistakes.
1. The resolution of figures are still low. It needs to improve as visable resolution.
2. There are still some space mistakes. ex) Double spaces in line 164, no space in line 621. Check them again.
Reviewer 3 Report
The Authors have addressed all my comments
The English language is improved. Check for spelling mistakes
Author Response
Please see the attachment。
